# Natural Compounds Attenuate Denervation-Induced Skeletal Muscle Atrophy

**DOI:** 10.3390/ijms22158310

**Published:** 2021-08-02

**Authors:** Tomohiko Shirakawa, Aki Miyawaki, Tatsuo Kawamoto, Shoichiro Kokabu

**Affiliations:** 1Division of Orofacial Functions and Orthodontics, Department of Health Improvement, Kyushu Dental University, Kitakyushu, Fukuoka 803-8580, Japan; r16shirakawa@fa.kyu-dent.ac.jp (T.S.); r15kawamoto@fa.kyu-dent.ac.jp (T.K.); 2Division of Molecular Signaling and Biochemistry, Department of Health Improvement, Kyushu Dental University, Kitakyushu, Fukuoka 803-8580, Japan; r17miyawaki@fa.kyu-dent.ac.jp

**Keywords:** skeletal muscle, maxillofacial muscle, atrophy, denervation, muscle homeostasis, natural compounds, royal jelly, geranylgeraniol

## Abstract

The weight of skeletal muscle accounts for approximately 40% of the whole weight in a healthy individual, and the normal metabolism and motor function of the muscle are indispensable for healthy life. In addition, the skeletal muscle of the maxillofacial region plays an important role not only in eating and swallowing, but also in communication, such as facial expressions and conversations. In recent years, skeletal muscle atrophy has received worldwide attention as a serious health problem. However, the mechanism of skeletal muscle atrophy that has been clarified at present is insufficient, and a therapeutic method against skeletal muscle atrophy has not been established. This review provides views on the importance of skeletal muscle in the maxillofacial region and explains the differences between skeletal muscles in the maxillofacial region and other regions. We summarize the findings to change in gene expression in muscle remodeling and emphasize the advantages and disadvantages of denervation-induced skeletal muscle atrophy model. Finally, we discuss the newly discovered beneficial effects of natural compounds on skeletal muscle atrophy.

## 1. Introduction

Skeletal muscle atrophy is the loss of the volume of skeletal muscle, leading to the weakness of muscle and causes disability. Skeletal muscle atrophy is known to be caused by immobility, aging, malnutrition, medication, or a wide spectrum of injuries or diseases that impact the nervous or musculoskeletal system. Sarcopenia was firstly described in 1989 as an age-related decrease in lean body affecting nutritional status, mobility, and independence [1]. Now, sarcopenia is defined as a progressive and generalized skeletal muscle disorder which involves the accelerated loss of muscle mass and function [2]. Thus, the definition of sarcopenia includes muscle loss related to physical inactivity, chronic disease, and malnutrition [3]. When age-related, it is known as primary sarcopenia. On the contrary, sarcopenia due to chronic disease or loss of mobility is called as secondary sarcopenia. However, this distinction is sometimes difficult because older patients sometimes present with both [4]. Sarcopenia is prevalent worldwide [5] and is recognized as a disease by the World Health Organization and included in the International Classification of Disease (ICD code M62.8) [6]. Therefore, skeletal muscle research has been actively conducted in recent years. However, therapeutic methods have not been established for skeletal muscle atrophy.

Sarcopenia has also occurred in the muscle of the head, neck, and maxillofacial region. Oral frailty, which impairs oral function, induces a high mortality rate [7]. The concept of oral frailty partially overlaps with sarcopenia of the muscle related to speaking, chewing, and swallowing [8]. In addition, facial muscle abnormalities cause facial and systemic disorders because skeletal muscles are involved in normal maxillofacial growth and malocclusion [9,10]. It is interesting to note that the developmental process and gene expression of facial muscles are different from that of other muscles [11,12,13].

Here, we summarize the muscle specificity in the maxillofacial region and emphasize the unique features of the maxillofacial region. We also explain the molecular mechanism of muscle development, anabolism, and catabolism. In addition, we focus on the sciatic nerve denervation model as a skeletal muscle atrophy model. Finally, we introduce foods that have been shown to be effective against skeletal muscle atrophy and discuss their usefulness.

## 2. Maxillofacial Problems Caused by Muscle Atrophy

Sarcopenia affects the maxillofacial region. Maxillofacial problems caused by muscle atrophy include oral frailty, malocclusion, and inhibition of normal growth and development. 

In recent years, attention has been paid to the decline in oral function known as oral frailty; muscle weakness in the maxillofacial region can affect dysphagia and communication, while oral frailty has also been reported to have a high mortality rate [7].

Maxillofacial muscles and dentition are closely related. The dentition is aligned in accordance with the muscle pressure exerted by the orbicularis oris, buccinator, and tongue muscles [10]. Thus, when muscle pressure imbalances occur, malocclusion occurs (Figure 1). 

For example, patients with muscular dystrophy have a lower tongue position than that seen commonly because of weakness of the lingual muscles. As a result, the maxillary arch narrows because of reduced tongue pressure. In addition, weakness of the masticatory muscles causes the mandible to lower downwards, resulting in supra-eruption of the molars. Therefore, it causes an anterior open bite.

Muscle weakness adversely affects the growth and development. Moss proposed functional matrix theory, which states that non-bone tissue induces bone growth in facial growth [9]. The growth of the membranous neurocranium and the naso-maxillary complex is regulated by environmental factors such as organ position and size, as well as the influence of soft tissues and teeth. Conversely, cartilage growth in the cartilage neurocranium, nasal septum, and mandibular condyle is highly regulated by genetic factors. A normal functional matrix is essential for obtaining a normal morphology. Craniofacial muscle weakness during growth hinders growth and development. In progressive facial hemiatrophy, muscle atrophy occurs unilaterally, resulting in facial asymmetry. Even after the growth is complete, deformation of the mandible can be caused by a mechanical imbalance in the masticatory muscles [14,15,16]. Moreover, masseter muscle removal [17] and masseter muscle damage [18] can cause mandibular asymmetry.

Muscles are also important factors in patients with jaw deformities. Patients with skeletal mandibular protrusion have low levels of myosin heavy chain (MyHC) expression [19], which may be attributed to the small occlusal contact points in patients with mandibular protrusion. Muscle effects have also been suggested to result in relapse after orthognathic surgery [20,21,22,23,24], however in contrast, it is interesting that the movement of bone fragments causes a change in the occupancy of the MyHC isoform. The masseter muscle usually has a high proportion of type I fibers; after orthognathic surgery, the proportion of type I fibers decreased and that of type II fibers increased [25,26,27]. Similarly, type I to type II conversions also occur at limb muscles due to severe deconditioning or spinal cord injury [28,29]. Decreased use of skeletal muscle has been reported in switching muscle fiber types from slow to fast [28], and it is difficult to determine whether it results from damage to the skeletal muscles, innervation caused by orthognathic surgery, or from intermaxillary fixation after orthognathic surgery. Interestingly, the gene expression level changes depending on the movement direction of the mandible (anterior or posterior movement) [20,21]. Muscle extension is thought to be caused by the movement of bone fragments.

Therefore, prevention and early treatment of muscle atrophy are required because muscle properties affect maxillofacial morphology and function. However, there are many unclear points about muscular atrophy in the maxillofacial region, and further research is needed.

## 3. Difference in Gene Expression between Facial and Other Muscles

In recent years, it has become clear that facial muscles and trunk muscles have significantly different properties even if they are the same type of muscle. In development, the facial muscles originate from the branchial arch, whereas the trunk muscles originate from the somites. In addition, the genes expressed in facial muscles and trunk muscles are different. Myogenic factor 5 (Myf5), myoblast determination protein 1 (MyoD), and paired box protein 7 (Pax7) are expressed in both muscles, but mesoderm posterior 1 (Mesp1) is found in facial muscles and Pax3 is found in trunk muscles [13]. However, expression of both Mesp1 and Pax3 has been confirmed in the muscles of tongue. Further, the gene expression of the facial muscles differs depending on the site, such as Myf5 in the eye muscles and Myf5/insulin gene enhancer protein (ISl1) in the masticatory muscles [30].

The reaction to muscle atrophy differs between facial and somatic muscles. It is known that in mature muscle tissue, facial muscles are less prone to cause muscle atrophy than somatic muscles in patients with muscular dystrophy. It had been thought that this is because the muscles of the face frequently contract to breathe and swallow. However, it was suggested that resistance to skeletal muscle atrophy may differ. Yoshioka et al. showed differences in muscle atrophy and regenerative ability between facial and trunk muscles using a skeletal muscle atrophy model [31]. In addition, the composition of the MyHC isoforms differs. It has been shown that expression of MyHC 2a is not observed in the masseter muscle in adult mice [32]. The characteristics of MyHC isoform are summarized in Table 1 [33,34,35].

In human masseter muscles, MyHC 2x are high expression. Interestingly, MyHC NEO is expressed in adult human masseter muscles [33]. In the trunk and limbs, MyHC NEO is usually expressed only during development and regeneration [33]. Satellite cells and muscle tissue stem cells are functionally heterogeneous populations in both masseter and limb muscles [36].

Due to differences in mechanism and characteristics, a site-specific approach may be required for muscle treatment. Research on the difference between facial muscles and trunk muscles is desired.

## 4. Molecular Mechanism of Muscle Homeostasis

### 4.1. Intracellular Signaling of Skeletal Muscle Anabolism

Insulin-like growth factor 1 (IGF-1) is mainly secreted from the liver and promotes the growth and proliferation of skeletal muscles. The mammalian target of rapamycin (mTOR) is a protein that plays a central role in the control of catabolism and anabolism, such as translation of mRNA, promotion of cell growth, and suppression of autophagy. There are rapamycin-sensitive mTOR complex 1 (mTORC1) that binds Raptor and rapamycin-non-sensitized mTOR complex 2 (mTORC2) that binds Rictor [37]. Protein kinase B (Akt) indirectly stimulates mTORC1. mTORC2 is required for phosphorylation of Akt [38]. Binding IGF-1 to the IGF-1 receptors leads to activation of mTORC1 via intracellular phosphoinositide 3-kinase (PI3K) and Akt. Conversely, when energy decreases, that is, when adenosine monophosphate (AMP) concentration increases, AMP-activated protein kinase (AMPK) is activated and protein synthesis is suppressed by negatively controlling mTORC1 activity [39]. mTORC1 promotes protein synthesis by suppressing eukaryotic translation initiation factor 4E (eIF4E)-binding protein 1 (4EBP1) and activating ribosomal protein S6 kinase-1 (S6K1), which regulate mRNA translation [40]. 4EBP1 inhibits eIF4E activity.

Satellite cells are the most abundant tissue stem cells resided in skeletal muscle; they are generally recognized for their contributions to regeneration, hypertrophy, and maintenance of muscle mass during the life span [41]. Pax7 is highly expressed in quiescent satellite cells, and MyoD expression is increased when cells are activated. Some satellite cells self-replicate and return to the quiescent phase, while others become myoblasts that express myogenin and cause muscle differentiation. Satellite cells are necessary for postnatal skeletal muscle growth, and trigger cytokines are secreted not only by inflammatory cells but also by muscle fibers, blood vessels, and motor neurons [42]. The trigger molecule has various factors such as IGF-1, interleukin-6 (IL-6), transforming growth factor β (TGF-β), and fibroblast growth factor (FGF), and is controlled intricately. IL-6 stimulation in skeletal muscle increases cyclin D1 expression via janus kinase 2 (JAK2)/signal transducer and activator of transcription 3 (STAT3) signaling, while IL-6, which is excessive due to chronic inflammation, suppresses satellite cell proliferation via JAK2/STAT3 signaling [43].

### 4.2. Intracellular Signaling of Skeletal Muscle Catabolism

Protein degradation system in muscle cells include the lysosomal system, the calpain system, and the ubiquitin-proteasome system. The ubiquitin-proteasome system consists of a ubiquitination system, which is comprised of ubiquitin activating enzyme (E1), ubiquitin binding enzyme (E2), and ubiquitin ligase (E3), and the 26S proteasome system that decomposes poly-ubiquitin [44]. The muscle-specific E3 ubiquitin ligases atrogin-1 and muscle ring finger 1 (MuRF-1) are upregulated in various muscle atrophy models. Forkhead box O (Foxo) is known as the gene that regulates the expression of atrogin-1 and MuRF-1. Foxo1 transgenic mice show decrease in muscle mass [45]. Atrogin-1 null mice and MuRF-1 null mice have been shown to suppress denervation-induced skeletal muscle atrophy [46]. 

Akt, which is activated by IGF-1, phosphorylates Foxo and suppresses the transcriptional activity of Foxo. Conversely, AMPK phosphorylates different sites of Foxo and increases the transcriptional activity [47]. The nuclear factor kappa-light-chain-enhancer of activated B cells (NF- κB), which is activated during inflammation, also promotes the induction of MuRF-1 expression [48]. Tumor necrosis factor α (TNF-α) also increases MuRF-1 expression via NFκB [49]. In addition, TNF-α increases Foxo activity by inhibiting the IGF1-Akt pathway via c-jun N-terminal kinase (JNK) [50].

Peroxisome proliferator-activated receptor gamma coactivator 1 (PGC1-α), which is a transcription conjugate factor, decreases when exercise is insufficient or inactive. A decrease in PGC1-α reduces phosphorylation of Foxo [51]. In addition, Casitas B-lineage lymphoma-b (Cbl-b) has been reported as a E3 ubiquitin ligase whose expression is highly increased in atrophic muscles. Cbl-b specifically binds to insulin receptor substrate-1 (IRS-1) to enhance ubiquitination and degradation, thus diminishing the IGF-1 signal [52,53]. It was confirmed that a peptide called Cblin, which is a Cbl-b inhibitor, inhibits the degradation of IRS-1 in the gastrocnemius muscle of mice undergoing sciatic nerve resection and suppresses the expression of muscle atrophy-related genes [53]. Cbl-b can be a treatment for skeletal muscle atrophy.

Myokines are secreted by skeletal muscles, some of which act on other organs and some of which act on the skeletal muscle itself. IGF-1, FGF-2, and IL-6 also act as myokines [54]. Myostatin, a member of the TGF-β superfamily, is secreted by skeletal muscle and has been reported as a factor that negatively regulates muscle growth [55]. Myostatin activates Smad2/3 via activin type II B receptors (ActRIIB) on the cell surface. Smad2/3 forms a heterodimer with Smad4. Smad2/3/4 regulates cell proliferation by controlling the expression of p21 and cyclin-dependent kinase 2 (CDK2). It also controls the expression of Pax7, MyoD, and myogenin to suppress muscle differentiation. In addition, phosphorylated Smad2/3 acts on Akt to inhibit mTORC1 activation and Foxo inactivation and negatively regulates protein levels [56].

## 5. Denervation Animal Model

The denervation model is a popular mechanism of skeletal muscle atrophy [57,58,59,60], where the sciatic nerve is usually removed. The reason is that nerves can be easily removed, surgery can be performed without damaging the tissue to be analyzed, and normal raising is possible after the surgery [61]. By analyzing the muscles (tibialis anterior muscle, gastrocnemius muscle, and extensor digitorum longus muscle) in the sciatic nerve innervation region, denervation-induced skeletal muscle atrophy is evaluated. Denervation-induced skeletal muscle atrophy upregulates the lysosome, calpain, and ubiquitin-proteasome systems [46,62,63]. The denervation model is useful for inducing major mechanisms of skeletal muscle catabolism. 

The advantage of the denervation model is that both an experimental group and a control group can be secured in the same animal [61]. By performing sciatic nerve denervation treatment on the experimental side and sham surgery on the opposite side, it is possible to ensure that muscles with and without atrophy can coexist in the same animal. It is beneficial for researchers to be able to eliminate the variability that occurs between individuals.

There are some things to consider in the denervation model. While skeletal muscle atrophy is induced by removing the sciatic nerve, there is a neuropathic pain model using the same method. According to the neuropathic pain model, it may cause self-harm on the affected limb, and it is necessary to observe excessive self-harm when evaluating the muscular atrophy model [64]. Further, the sciatic nerve denervation model is used as an osteoporosis model [65]. The prevalence of sarcopenia and osteoporosis has been reported to be correlated, and it has been suggested that myokine affects bone [66]. In recent years, it has been suggested that osteokine secreted by bone tissue affects the whole body [54]. These reports mean that the denervation-induced skeletal muscle atrophy model does not merely reflect skeletal muscle atrophy; it suggests that it is affected by neural transmission, trophic substances, and secretions from bone tissue.

Moreover, not all mechanisms of muscle atrophy can be elucidated by the denervation model. Other animal models include the hind limb unloading and immobilization models, while the hind limb unloading model induces skeletal muscle atrophy by lifting the legs. This model induces skeletal muscle atrophy similar to the microgravity-like space area. The immobilization model imitates bedrest and induces muscle atrophy in immobile legs using casts. These models differ in the mode of skeletal muscle atrophy. For example, in the space area, the rat soleus muscle undergoes severe atrophy, but the tibialis anterior muscle shows less atrophy [67]. In the immobilization model, protein degradation by lysosomes was lower than that in the other models [68]. These reports differ from that of denervation-induced skeletal muscle atrophy. The use of other models is considered for skeletal muscle atrophy under these specific conditions. 

While some molecular mechanisms are common among skeletal muscle atrophy models, there are specific mechanisms to denervation-induced skeletal muscle atrophy. Activation of mTORC by denervation stimulates S6K, resulting in suppression of IRS-1 as a negative feedback effect, and it has been suggested that Foxo-upregulation occurs as a result [69]. In addition, it has been shown that muscle atrophy occurs even if myostatin is inhibited during denervation, and muscle fibers do not recover. In contrast, the immobilization model has been shown to suppress muscle atrophy by inhibiting myostatin, which shows that there are differences in the mechanism of muscle atrophy models [57]. Further detailed molecular mechanisms are expected to be elucidated in future.

Furthermore, attention should be paid to skeletal muscle atrophy in the craniofacial region. As mentioned earlier, the development pattern is different between the facial and limb muscles. The limb muscles are controlled by the motor nerves from the spinal cord, while the facial muscles are controlled by the cranial nerves. The orbicularis oris and buccinator muscles are controlled by the facial nerve (cranial nerve VII), and facial nerve axotomy models have been established [70]; however, studies on facial muscle atrophy are insufficient. In the future, it is necessary to investigate whether gene expression differs from that of somatic muscle during muscle atrophy of facial muscles.

The denervation model is considered to be the following disease model. Spinal muscular atrophy, amyotrophic lateral sclerosis, and neuralgic amyotrophy are caused by motor nerve degeneration, and because these diseases impair facial muscle movement and swallowing, the sciatic nerve denervation model may be useful for elucidating the mechanism of muscular atrophy in maxillofacial region. These diseases indicate a poor prognosis. Motor nerve degeneration or denervation occurs not only in congenital diseases but also in acquired factors (for example, injuries, virus infection, and surgery). Denervation is thought to contribute to sarcopenia, because motor innervation of skeletal muscle decreases with aging [71]. Sarcopenia affects other diseases, with osteoporosis [72,73], bone fractures [74], critical limb ischemia [75], diabetes [76], cognitive decline [77], and cancer [78,79,80] showing a higher predilection in sarcopenia. In other words, an increase in skeletal muscle mass is important for the prevention of various diseases. Therapy and prevention of skeletal muscle atrophy need to intervene the catabolic downregulation and/or anabolic upregulation. Currently, however, there are no therapeutic strategies to approach the muscle remodeling cycle (Figure 2). 

This schema is muscle remodeling cycle. Increasing muscle anabolism (proliferation, differentiation, muscle hypertrophy) and reducing catabolism are necessary against skeletal muscle atrophy. Satellite cells are stimulated by damaged-myofiber-derived factors from the muscle tissue [81]. This cycle is also affected by hormones, myokines, osteokines, and adipokines.

## 6. Natural Compounds (Effective Foods against Denervation-Induced Skeletal Muscle Atrophy)

There are a wide variety of pathologies that cause skeletal muscle atrophy. However, many factors, such as hereditary diseases and sarcopenia, are currently difficult to eliminate. In addition, the muscle remodeling cycle is fast, and long-term treatment is necessary to maintain muscle mass. Therefore, if skeletal muscle atrophy can be prevented by food, it would be an effective population approach. These diets are safe, inexpensive, and can be incorporated into daily intake. In addition, since food can cause allergies, it is necessary to find as many foods as possible that can counteract skeletal muscle atrophy.

The following is a summary of effective foods against denervation-induced skeletal muscle atrophy.

### 6.1. Royal Jelly (RJ)

Honeybees (e.g., *Apis mellifera*) excrete RJ from cephalic glands. RJ is the main source of nutrition for queen honeybees; they are larger, with a longer life span than other honeybees, while RJ affects the fertility of queen honeybees [82,83]. It has been reported to prolong life span [84,85], reduce fatigue [86], and have antioxidant and anti-inflammatory properties [87,88,89]. In humans, RJ reduces serum cholesterol and lipid levels [90].

The components of RJ include water (60–70%), proteins (9–18%), sugars (7.5–23%), lipids (3–8%), and other trace compounds. RJ contains 60–80% trans-10-hydroxy-2-decenoic acid (10H2DA) and 10-hydroxydecanoic acid (10HDAA) in lipids [91]. In animal experiments, 10H2DA and 10HDAA were found to be pharmacologically beneficial [92,93,94,95,96].

RJ affects skeletal muscle metabolism. In mice experiments, RJ induces regeneration of damaged skeletal muscle by satellite cells via the IGF-1-Akt pathway [97] and activation of AMPK by endurance training [98]. RJ removed protein (protease-treated RJ [pRJ]) also had a positive effect on skeletal muscle. Daily oral administration of pRJ prevents denervation-induced skeletal muscle atrophy [99], and it has been reported that pRJ affects muscle fiber thickness, expression of satellite cell catabolic gene, and proliferation and differentiation in C2C12 myoblasts [99,100]. 

Although it is known that RJ suppresses skeletal muscle atrophy, a detailed mechanism has not been revealed. RJ is known to regulate epigenetic changes [101]. RJ and 10H2DA suppressed histone deacetylase (HDAC)-activity without affecting DNA methylation [102]. Inhibition of HDAC and DNA methyltransferases upregulates myogenesis [103,104,105,106]. The epigenetic effects of RJ should be further investigated and need to be examined for changes in gene expression.

RJ upregulates IGF-1, IGF receptors, and pAMPK. Activation of Akt and AMPK translocate glucose transporter type 4 (GLUT4) to the cell membrane [107]. 10H2DA, an RJ-specific fatty acid, activates AMPK in skeletal muscles [108]. Mitochondrial activity in skeletal muscle is related to insulin resistance and is important for preventing sarcopenia. These studies suggest a therapeutic approach to glucose tolerance with decreased skeletal muscle.

The beneficial effects of RJ have only been partially elucidated. In future, a detailed downstream analysis of RJ-specific components is required.

### 6.2. Geranylgeraniol (GGOH)

GGOH is a C20 isoprenoid found in fruits, vegetables, and grains. GGOH falls under the category of “Generally recognized as safe (GRAS)” for consumption [109]. GGOH is an intermediate product of the mevalonate pathway and functions as a precursor of geranylgeranylpyrophosphate (GGPP). 

Matsubara et al. showed that GGOH enhances C2C12 myoblast differentiation in vitro, but high doses of GGOH tend to suppress myoblast proliferation [110]. Miyawaki et al. reported that GGOH administration increased the muscle fiber size in denervation-induced skeletal muscle atrophy in vivo [111]. GGOH also suppresses the denervation-induced or glucocorticoid-induced atrogin-1 expression [111]. Expression of atrogin-1 is increased when muscle atrophy is induced by the stressors [112]. Suppressing atrogin-1 expression is important to prevent skeletal muscle atrophy. 

Many studies have shown the role of NF- κB in the induction of muscle atrophy [48,113,114,115,116,117]. NF- κB upregulates atrogin-1 expression [118]. GGOH treatment decreases lipopolysaccharide (LPS)-induced NF- κB signaling [119,120]. GGOH has also been demonstrated to upregulate testosterone synthesis in testis-derived cells [121]. Testosterone is a steroid hormone that is strongly involved in muscle metabolism [122]. Androgen and testosterone promote muscle hypertrophy and suppress the expression of atrogin-1 and MuRF-1 [123,124]. Therefore, NF- κB signaling and/or testosterone may participate in the suppression of skeletal muscle atrophy by GGOH.

Statins are used to prevent cardiovascular disease [125,126,127,128] and inhibit cholesterol synthesis via the mevalonate pathway. However, they may induce muscle cell damage and severe rhabdomyolysis [129,130,131,132]. Statin-associated muscle disorders may reduce crucial intermediary molecules such as GGPP by inhibiting the mevalonate pathway [133,134,135]. Treatment of C2C12 cells with GGPP reverses the inhibitory effect of statins on myotube formation [136]. Cao P et al. reported that GGOH treatment reduces the expression levels of atrogin-1 that is induced by statins in vitro [137].

GGOH is inexpensive, classified as GRAS, and can be administered orally. In future, a detailed downstream analysis of GGOH is required on skeletal muscle metabolism.

### 6.3. Soybeans

Soybeans are grown in many countries for food, fertilizer, and oil production. The components of soybeans include proteins (33.8%), sugars (29.5%), lipids (19.7%), water (12.4%), and other trace compounds [138]. Soy protein has been reported to promote increased skeletal muscle mass and strength in humans [139].

Long-term administration of soy protein increased the number of satellite cells and differentiated cells (pax7^−^ myoD^+^) in ovariectomized mice [140]. Glycinin is a major protein contained in soybeans [141]. Glycinin has an amino acid sequence similar to that of Cblin. When the muscle atrophy inhibitory effect of glycinin in denervation mice was examined, it was confirmed that it suppressed the decrease in the muscle wet weight of the tibialis anterior muscle and suppressed the decrease in muscle cross-sectional area [142]. A mixed diet of soy protein and whey protein showed a strong inhibitory effect on denervation-induced skeletal muscle atrophy [143]. In addition, it has been reported that soy protein isolate and red bell pepper juice suppressed skeletal muscle atrophy in denervated mice [144]. Whey protein is known to stimulate muscle protein synthesis via mTOR signaling in humans [145]. It has been reported that ingestion of soy protein also increases muscle mass in human with low physical activity [139].

Soybeans are nutritious and contain the isoflavones described in the next section. It is necessary to determine the components that act on the skeletal muscles.

### 6.4. Polyphenol

Polyphenol is a compound containing multiples of phenol units contained in plants. Various uses of polyphenols have been reported, including effective substances on muscle atrophy in recent years.

Isoflavones are one of the flavonoids and are natural organic compounds. Administration of isoflavones suppresses denervation-induced apoptosis and muscle atrophy [146,147]. It has been reported that isoflavones can suppress the transcriptional activity of MuRF-1 induced by TNF-α and myotube atrophy [148]. In addition, isoflavones suppress the damage of acetylcholine receptors through denervation. It has been suggested that they have a protective effect on neuromuscular junctions [147].

Isoflavones are known to act as phytoestrogens. Women who consumed isoflavones for 24 weeks had an increased muscle mass index [149]. Estrogen receptors (ER) include ERα and ERβ. Daidzein, a soy isoflavone, has been shown to have more effect on ERβ than on Erα [150]. ERβ is involved in the synthesis and degradation of skeletal muscles. It has been shown to promote muscle fiber growth via Erβ [151]. In addition, it was clarified that it is responsible for the inhibition of satellite cell proliferation and cell death via Erβ [151]. A decrease in the blood levels of sex hormones that occur in old age leads to a decrease in skeletal muscle. Daidzein may be effective in older women, and in addition, 8-prenylnaringenin, which has an estrogenic effect like daidzein, suppresses denervation-induced skeletal muscle atrophy, and its effect is thought to be due to phosphorylation of Akt [152]. These estrogenic isoflavones may be effective not only for skeletal muscle atrophy, but also for postmenopausal disease.

Quercetin is a flavonoid contained in fruits and vegetables, and has a strong radical scavenging ability. Reactive oxygen species (ROS) are generated during the process of ATP production in mitochondria. ROS causes cell damage and is involved in the disease [153,154,155]. ROS activates the NFκB and Foxo pathways and induces the expression of E3 ubiquitin ligase [156,157]. Research is ongoing on the concept that substances with antioxidant activity suppress ROS and muscle atrophy. In the hind limb unloading model, administration of quercetin to the gastrocnemius muscle reduced atrogin-1 and MuRF-1 expression and suppressed the loss of skeletal muscle mass [158]. Administration of quercetin promotes phosphorylation of Akt and suppresses skeletal muscle atrophy [159].

### 6.5. Vitamins

Vitamins are nutrients that cannot be synthesized in sufficient amounts in the body, and are organic compounds excluding the three major nutrients. Insufficient vitamin intake has systemic effects but is also an important factor in skeletal muscle. 

Vitamin C is a cofactor involved in the synthesis of collagen. Vitamin C is usually considered to have an antioxidant effect and eliminates ROS [160]. However, some reports show that vitamin C have prooxidant effect [161,162]. It was reported that elderly women with high concentration of vitamin C had high muscle strength and physical ability [163]. When genetically modified mice that could not biosynthesize vitamin C were used, vitamin C deficiency reduced muscle weight and increased expression of Foxo-1, Cbl-b, atrogin-1, and MuRF-1. Further, re-administration of vitamin C rescued muscle weight and reduced skeletal muscle atrophy gene expression [164]. Conversely, Makanae et al. reported that the supplementation of vitamin C suppresses the hypertrophy of muscle by overloading [165]. Like that, the function of vitamin C in muscle metabolism is controversial and may be different depending on the physical status.

Vitamin D is involved in calcium absorption and utilization and bone calcification, but in recent years it has been suggested that it may also play an important role in skeletal muscle [166]. In mice, deletion of vitamin D receptor reduces muscle fiber size [167]. Low levels of vitamin D in the blood reduce muscle strength and increase the risk of sarcopenia [168]. There is also a report that vitamin D administration restores muscle strength [169]. In experiments using C2C12 cells, vitamin D suppressed the expression of atrogin-1 and cathepsin L [170]. These literatures also showed that the supplementation of vitamin D is effective only in the condition of vitamin D deficiency.

Vitamin E has an antioxidant effect similar to that of vitamin C; therefore, it is expected to remove ROS. In the Unload model, vitamin E reduced the expression of atrogin-1 and MuRF-1 and suppressed skeletal muscle atrophy [171]. In this study, muscular atrophy suppression is not due to an antioxidant effect. In contrast, there are reports that vitamin E has no suppressive effect on muscular atrophy [172]. 

### 6.6. Capsaicin

Capsaicin is a pungent ingredient contained in chili peppers. The capsaicin receptor transient receptor potential vanilloid 1 (TRPV1) is known as a pain receptor. It was reported that TRPV1-mediated Ca^2+^ signaling activates mTOR and promotes muscle hypertrophy [173]. However, the stimulation of TRPV1 causes pain; therefore, it seems difficult to apply it as a food approach.

Figure 3 shows how natural compounds are effective in muscle remodeling cycle and Table 2 summarizes the studies to examine the effect of natural compounds on skeletal muscle atrophy in human and animal model. Unfortunately, no studies have reported the function of natural compounds on maxillofacial muscle atrophy and sarcopenia. Special attention and research on maxillofacial muscle should be required immediately. Because the malnutrition by the problems of occlusion and/or swallowing may contribute to the loss of whole body muscles. 

These are natural compounds that suppress denervation-induced skeletal muscle atrophy. In other words, they induce the increasing muscle anabolism and/or decreasing muscle catabolism. The mechanism of suppressing skeletal muscle atrophy differs depending on the foods. RJ: Royal jelly, GGOH: Geranylgeraniol.

**Table 2 ijms-22-08310-t002:** Studies for the function of natural compounds on skeletal muscle metabolism.

References	Species	Natural Compounds	Phenotype, Intervention/ Key Findings
Niu K et al. [97]	mice	Royal jelly (RJ)	C57BL/6J mice, aged mice/ Suppression of decrease muscle weight and grip strength Increase the regeneration of injured muscles and the serum insulin-like growth factor-1 (IGF-1)
Takahashi Y et al. [98]	mice	RJ	ICR mice, training/ RJ induces mitochondrial adaptation with endurance training by AMP-activated protein kinase (AMPK) activation
Shirakawa T et al. [99]	mice	RJ	C57BL/6J mice, denervation/ Suppression decrease muscle fiber size by oral administration
Okumura N et al. [100]	mice	RJ	Genetically heterogeneous mice, aged mice/ Motor function Increase fiber size Increase proliferation and differentiation
Takikawa M et al. [108]	mice	10H2DA	C57BL/6J mice, oral adnimistration/ Stimulated phosphorylation of AMPK Glucose transporter type 4 (Glut4) translocation to the plasma membrane
Miyawaki A et al. [111]	mice	Geranyl-geraniol (GGOH)	C57BL/6J mice, denervation/ Suppression decrease muscle fiber size and expression of atrogin-1
Hashimoto R et al. [139]	Humans	Soy protein	High and low physical activity, food intake/ Increase skeletal muscle mass in low activity human
Kitajima Y et al. [140]	mice	soymilk	C57BL/6 mice, ovariectomized mice/ Muscle fiber hypertrophy Increase grip strength
Abe T et al. [142]	mice	Soy glycinin	C57BL/6J mice, denervation/ Increase fiber diameter Suppression expression of muscle atrogene via IGF-1 signaling
Nikawa T et al. [143]	mice	Soy protein and whey protein	C57BL/6 mice, denervation/ Suppression muscle atrophy
Tachibana N et al. [144]	mice	Soy protein and red bell pepper juice	C57BL/6J mice, denervation/ Suppression of muscle atrophy and decrease atrogenes
Kakigi R et al. [145]	Humans	Whey protein	Male, Food intake, resistance exercise/ Mammalian target of rapamycin (mTOR) signaling activate
Tabata S et al. [146]	mice	isoflavones	ICR mice, denervation/ Suppression muscle atrophy Decrease in apoptosis-dependent signaling
Hirasaka K et al. [147]	mice	Soy isoflavones	C57BL/6J mice, denervation/ Resistance to muscle atrophy Suppression of acetylcholine receptor disorders in denervating atrophic muscles
Aubertin-Leheudre M et al. [149]	Humans	isoflavones	Sarcopenic-obese women, food intake/ Increase fat-free mass and muscle mass index
Mukai R et al. [152]	mice	8-Prenylnarin-genin	C57/BL6 mice, denervation/suppress muscle atrophy Increased phosphorylation of Akt Suppression expression of Atrogin-1
Mukai R et al. [158]	mice	quercetin	C57BL/6J mice, tail suspension/ Suppression decrease muscle weight and express ubiquitin ligase
Mukai R et al. [159]	mice	quercetin	C57BL/6 mice, denervation/ Suppression muscle atrophy Decrease Reactive oxygen species (ROS) Increased phosphorylation of Akt
Saito K et al. [163]	Humans	Vitamin C	Women, 70-84 years old/ Plasma vitamin C levels are positively correlated with grip strength, length of time standing on one leg with eyes open, and walking speed
Takisawa S et al. [164]	mice	Vitamin C	SPM30 knockout mice/ Muscle atrophy due to vitamin C deficiency, and recovery of muscle mass after vitamin C supplementation
Makanae Y et al. [165]	rats	Vitamin C	Wistar rats, overload/ suppression muscle hypertrophy on overload by administration of vitamin C
Ceglia L et al. [166]	Humans	Vitamin D	Mobility-limited, vitamin D-insufficient women/ Increase muscle fiber by supplemental vitamin D
Endo I et al. [167]	mice	Vitamin D	Vitamin D receptor (VDR) deletion mice/ Muscle fiber contraction by deletion of VDR
Visser M et al. [168]	Humans	Vitamin D	55–85 years old/ In humans with low serum vitamin D, lower grip test and tend to low appendicular skeletal muscle mass
Servais S et al. [171]	rats	Vitamin E	Wistar rats, hindlimb-suspend/ Suppression of muscle atrophy and decrease atrogenes
Ikemoto M et al. [172]	rats	Vitamin E	Wistar rats, tail suspension/ Supplemental vitamin E does not show effect of suppression muscle atrophy
Ito N et al. [173]	mice	capsaicin	Denervation, hindlimb suspension/ Suppression of muscle atrophy by capsaicin injected intramuscularly

The table is summarized the literatures to investigate the function of natural compounds skeletal muscle metabolism.

## 7. Conclusions

The dynamics of muscle atrophy are complex and diverse. If food is effective in preventing skeletal muscle atrophy, it is safe and applicable to many people. The substances introduced this time can be candidates for treatment methods, but we think there are other beneficial substances. Animal models, including denervation models, are effective for elucidating molecular mechanisms and developing therapeutic substances. Further research focusing on foods against skeletal muscle atrophy is needed to study the mechanism of skeletal muscle atrophy.

## Figures and Tables

**Figure 1 ijms-22-08310-f001:**
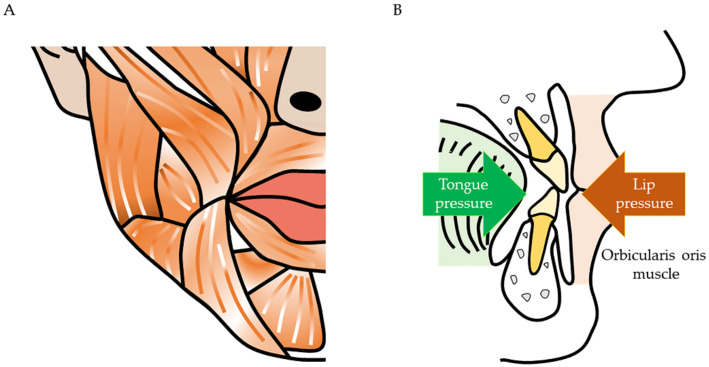
Relationship between dentition and facial muscles. (**A**) The muscles run vertically and horizontally on the face. (**B**) The teeth move to the position where the pressure is balanced. The dentition receives muscle pressure medially from the tongue muscles and laterally from the orbicularis oris and buccinator muscles. Orthodontists perform myofunctional therapy because abnormal oral habits (e.g., infant-type swallowing and tongue thrusting habit) adversely affect the dentition.

**Figure 2 ijms-22-08310-f002:**
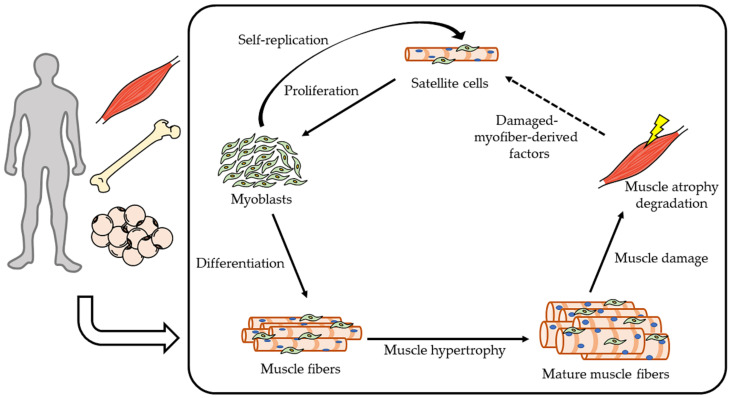
Muscle remodeling cycle.

**Figure 3 ijms-22-08310-f003:**
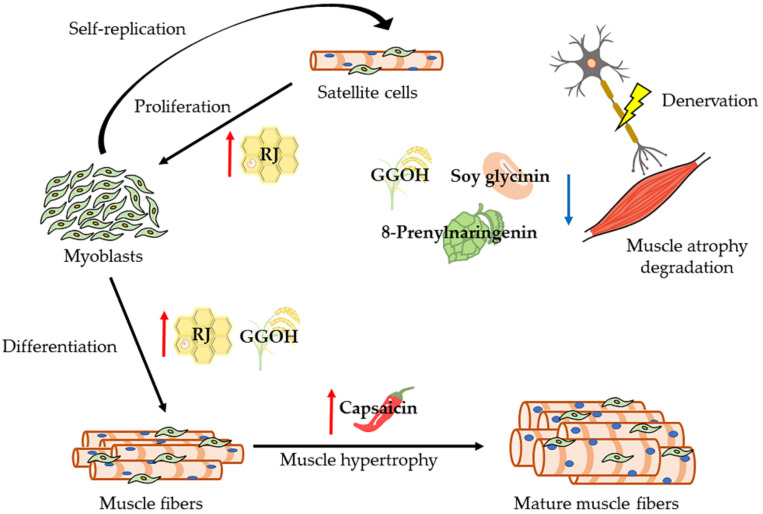
Schematic overview of the effective natural compounds in muscle remodeling cycle.

**Table 1 ijms-22-08310-t001:** Comparison of myosin heavy chain isoform.

Genes	Proteins	Characteristics
*MYH 1*	MyHC 2x	Fast type 2x fibers
*MYH 2*	MyHC 2a	Fast type 2a fibers
*MYH 3*	MyHC EMB	Developing muscle, Extraocular muscles
*MYH 4*	MyHC 2b	Fast type 2b fibers
*MYH 6*	MyHC α	Heart and jaw muscles
*MYH 7*	MyHC β	Heart and slow muscles, type 1 fibers
*MYH 7b*	MyHC slow tonic	Extraocular muscles
*MYH 8*	MyHC NEO	Developing muscle, expression in masseter muscles
*MYH 13*	MyHC EO	Extraocular muscles
*MYH 15*	MyHC 15	Extraocular muscles
*MYH 16*	MyHC 16	Jaw muscles (in human, translation is blocked)

Each Myosin heavy chain (MyHC) isoform has own characteristics and is different to expression depending on the location and timing of muscles.

## Data Availability

Not applicable.

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
