# Peer review of "Natural Compounds Attenuate Denervation-Induced Skeletal Muscle Atrophy"

_ijms, 2021, doi:10.3390/ijms22158310_

Round 1

Reviewer 1 Report

Shirakawa et al. provide an interesting review that focuses on muscle atrophy in special skeletal muscle groups and effect of natural compounds with the potential to treat muscle atrophy due to various causes. While the review addresses these issues from an original perspective, there are several deficiencies that need to be addressed.

Comments:

  1. More precise definition of sarcopenia should be used.
  2. While sarcopenia is often used to describe age-related muscle loss, this is not an official classification, which distinguishes primary and secondary sarcopenia. The paper would gain if the classification with appropriate references is at least mentioned.
  3. The difference between sarcopenia and muscle atrophy should be mentioned.
  4. The introductory part focuses first on sarcopenia and then on the problems of atrophy of skeletal muscles in the maxillofacial region. However, the link between the two is not sufficiently described. For instance, to what extent are muscles in this region atrophic in persons with sarcopenia?
  5. While the section focuses on cranial skeletal muscles, transitions of MyHC refer to spinal cord injury, which would affect limb muscles (l. 84 and 85). It would be better if limb muscles are mentioned directly in this context.
  6. The relevance of the sciatic nerve lesions to the discussion about the atrophy of muscles in the maxillofacial region is not sufficiently explained.
  7. Rapamycin-sensitized should probably be rapamycin-sensitive.
  8. Figure 2 appears to suggest that muscle hypertrophy occurs by addition of new muscle fibers (hyperplasia) rather than by hypertrophy of muscle fibers that already exist.
  9. Differences in the expression of MyHC in different muscle groups should be more detailed and precise.
  10. Sections on muscle metabolism are mainly describing pathways involved in protein synthesis rather than metabolism in general. A more appropriate section title should therefore be chosen.
  11. Myokines are a group of molecules (i.e. it is better not to speak about myokine as being a factor).
  12. The role of satellite cells in the context of muscle hypertrophy should be more clearly explained.
  13. Some parts of the text, such as introductory paragraphs in the subsection 5 (denervation animal models), require references. Several claims are provided without any references.
  14. The part that describes possible therapeutic compounds should cite more human studies that examined effectiveness of vitamins and other agents for treatment of muscle wasting. This part of the text should also relate more directly to the issue of muscle atrophy in the maxillofacial region (if this is the topic of the review).
  15. Discussion about the use of vitamins should mention that vitamin C may have pro-oxidant effects. Moreover, vitamin C supplementation may block beneficial effects of exercise, which should also be mentioned.
  16. Discussion about the vitamins should also distinguish between their use for vitamin deficiencies as opposed to the use of vitamins in persons who have normal vitamin levels. For instance, vitamin D may be beneficial in those with vitamin D deficiency, but not in those who have normal levels of vitamin D.
  17. It would be useful to include tables, which provide an overview of studies, which examined effects of natural compounds on muscle mass in different models, including human studies.
  18. TRPV1 is misspelled TRAPV1
  19. It is not clear what is meant by muscle joints.

Author Response

Thank you very much for your kind comments regarding our manuscript. Your remarks have aided us to significantly improve the manuscript. We have considered your constructive comments and have revised the manuscript by adding new information. The changes are marked with Red colored text.

Comment 1: More precise definition of sarcopenia should be used.

In accordance with your comments, we have added the sentences which included the precise definition of sarcopenia at the introduction section (at Line 29 - 34) in the revised manuscript.

Comment 2: While sarcopenia is often used to describe age-related muscle loss, this is not an official classification, which distinguishes primary and secondary sarcopenia. The paper would gain if the classification with appropriate references is at least mentioned.

In accordance with your comments, we have added classification of sarcopenia with appropriate references at the introduction section (at Line 34 - 37) in the revised manuscript.

Comment 3: The difference between sarcopenia and muscle atrophy should be mentioned.

In accordance with your comments, we have had added sentences including the difference between sarcopenia and other muscle atrophy at the introduction section (at Line 26 - 34) in the revised manuscript.

Comment 4: The introductory part focuses first on sarcopenia and then on the problems of atrophy of skeletal muscles in the maxillofacial region. However, the link between the two is not sufficiently described. For instance, to what extent are muscles in this region atrophic in persons with sarcopenia?

In accordance with your comments, we have had added the sentences linking between sarcopenia and the problems of atrophy of skeletal muscle in the maxillofacial region at the introduction section (at Line 42 - 47) in the revised manuscript.

Comment 5: While the section focuses on cranial skeletal muscles, transitions of MyHC refer to spinal cord injury, which would affect limb muscles (l. 84 and 85). It would be better if limb muscles are mentioned directly in this context.

In accordance with your comments, we have added the explanation and emphasis at limb muscles, the transition of type of MyHC occurs due to severe deconditioning or spinal cord injury at Line 97- 99 in the revised manuscript.

Comment 6: The relevance of the sciatic nerve lesions to the discussion about the atrophy of muscles in the maxillofacial region is not sufficiently explained.

Thank you for your very constrictive comments. However, unfortunately, we could not find the literatures to explain the relevance between sciatic nerve injury and muscle atrophy in the maxillofacial region. Therefore, we are so sorry, but we cannot remark it in this review.

Comment 7: Rapamycin-sensitized should probably be rapamycin-sensitive.

We apologize for the typos and have corrected this at the Line 149 in the revised manuscript.

Comment 8: Figure 2 appears to suggest that muscle hypertrophy occurs by addition of new muscle fibers (hyperplasia) rather than by hypertrophy of muscle fibers that already exist.

Thank you for your very constrictive comments. we have realized that our original figure 2 is not correct as the explanation of hypertrophy. So, in accordance with your comments, we have modified figure 2 and 3 to explain hypertrophy precisely, that is, muscle fibers already existed increases their size.

Comment 9: Differences in the expression of MyHC in different muscle groups should be more detailed and precise.

We agree with your comment. in accordance with your comment, we have added some sentences at Line 128 - 137, and table 1 to explain these differences more detailed and precise in the revised manuscript.

Comment 10: Sections on muscle metabolism are mainly describing pathways involved in protein synthesis rather than metabolism in general. A more appropriate section title should therefore be chosen.

In accordance with your comment, we have modified the section title at Line 143 in the revised manuscript.

Comment 11: Myokines are a group of molecules (i.e. it is better not to speak about myokine as being a factor).

In accordance with your comment, we have removed the words “hormonal factor” from the sentence (Line 202) to precisely describe myokines in the revised manuscript.

Comment 12: The role of satellite cells in the context of muscle hypertrophy should be more clearly explained.

In accordance with your comments, we have added the explanation of the role of satellite cells in the context of muscle hypertrophy at the section of “intracellular signaling of skeletal muscle anabolism” in the revised manuscript.

Comment 13: Some parts of the text, such as introductory paragraphs in the subsection 5 (denervation animal models), require references. Several claims are provided without any references.

In accordance with your comment, we have added the adequate references at the subsection 5 (at Line 214 - 228) in the revised manuscript. In addition, through this manuscript, we have checked the references. So, we have also added some references at Line 179 and 187 - 194 though the revised manuscript.

Comment 14-1: The part that describes possible therapeutic compounds should cite more human studies that examined effectiveness of vitamins and other agents for treatment of muscle wasting.

In accordance with your comment, we have added the information of humans studied and combined these with the studies described in original manuscript as table 2 in the revised manuscript.

Comment 14-2: This part of the text should also relate more directly to the issue of muscle atrophy in the maxillofacial region (if this is the topic of the review).

We completely agree with your comments. However, unfortunately, we could not find the literatures to show the effectiveness of natural compounds on the muscle atrophy in the maxillofacial region. Therefore, we are so sorry, but we cannot remark it in this review.

Comment 15: Discussion about the use of vitamins should mention that vitamin C may have pro-oxidant effects. Moreover, vitamin C supplementation may block beneficial effects of exercise, which should also be mentioned.

In accordance with your comment, we have added the description of pro-oxidant effect at Line 413 - 415 and blocking beneficial effects of exercise at Line 420 - 423 in the revised manuscript.

Comment 16: Discussion about the vitamins should also distinguish between their use for vitamin deficiencies as opposed to the use of vitamins in persons who have normal vitamin levels. For instance, vitamin D may be beneficial in those with vitamin D deficiency, but not in those who have normal levels of vitamin D.

In accordance with your comment, we have added the explanation that the supplementation of vitamin D is effective only in the condition of vitamin D deficiency at Line 430 - 431 in the revised manuscript.

Comment 17: It would be useful to include tables, which provide an overview of studies, which examined effects of natural compounds on muscle mass in different models, including human studies.

Thank you for your very constrictive comments. In accordance with your comment, we have added the table to provide overview of studies, which examined effects of natural compounds on muscle mass, as table 2 in the reversed manuscript.

Comment 18: TRPV1 is misspelled TRAPV1.

We apologize for the typos and have corrected this at Line 440 - 441 in the revised manuscript.

Comment 19: It is not clear what is meant by muscle joints.

We apologize for the usage of unclear word “muscle joints”. We have replaced this with “neuromuscular junction” at Line 387 in the revised manuscript.

We sincerely hope that our efforts to address all the concerns of the reviewers, which include new figures, and major revisions to the manuscript, make this work suitable for publication.  

Reviewer 2 Report

This is a well-written review with different sections focusing on various sections. The interesting sections of this review were the authors' focus on the importance of skeletal muscle in the maxillofacial region and highlight the differences between skeletal muscles in the maxillofacial region and other regions. Another section that I found very interesting and useful is discussing the foods that have been shown to be effective against skeletal muscle atrophy. I believe this work is very beneficial for the readers interested in skeletal muscle and food science. 

Author Response

We are thankful for the time and energy you expended. We appreciate that you regarded current form of our manuscript as acceptable. However, we have added some modification to the reversed manuscript according to the other reviewer's comments. We hope you will understand this situation. 

We sincerely hope that our efforts to address all the concerns of the reviewers, which include new figures, and major revisions to the manuscript, make this work suitable for publication.  

Round 2

Reviewer 1 Report

The authors have thoroughly revised the manuscript. The revision of figure 2 and inclusion of tables is particularly appreciated.

Comments:

  • The title indicates that natural compounds prevent atrophy by denervation. However, while the revised manuscript includes more human data, the level of evidence that natural compounds actually prevent muscle atrophy due to denervation is not extremely strong. I therefore suggest to the authors to consider a more balanced title, which takes into account that most effects of natural compounds are modulatory rather than preventive.
  • Genes encoding MyHC proteins are more commonly abbreviated as MYH and not as MyHC. I therefore suggest to the authors that they should use the commonly accepted names of these genes (in italics) MYH1, MYH2,…
  • Since the focus is on maxillofacial region, a short comment regarding the sciatic nerve denervation model and atrophy/sarcopenia in the maxillofacial region would be beneficial for the paper. For instance, is the denervation model perhaps (at least indirectly) relevant for diseases, such as amyotrophic lateral sclerosis, which affects the upper and lower motor neuron and causes problems with swallowing (in addition to problems with movement)?
  • I suggest to the authors to say “myokines are secreted” instead of “myokine is secreted”. (Myokines are usually referred to as a group of molecules.)
  • If there are no data regarding the effects of natural compounds on muscle atrophy/sarcopenia in the maxillofacial region, the authors could mention directly that this aspect should be studied in the future. This is relevant also because persons with swallowing problems may have insufficient intake of macro- and micronutrients, such as vitamins, which may contribute to the loss of muscles. This would be a nice conclusion of this review as well as a perspective for future research.
  • Chorionic disease should be chronic disease (line 36). There are a few other minor errors like this (e.g. vitamin C have instead of vitamin C has…in line 426). I suggest a careful proofreading of the manuscript.

Author Response

To Reviewer

Thank you very much for your kind comments regarding our manuscript. Your remarks have aided us to significantly improve the manuscript. We have considered your constructive comments and have revised the manuscript by adding new information. The changes are marked with Brue colored text.

#1. The title indicates that natural compounds prevent atrophy by denervation. However, while the revised manuscript includes more human data, the level of evidence that natural compounds actually prevent muscle atrophy due to denervation is not extremely strong. I therefore suggest to the authors to consider a more balanced title, which takes into account that most effects of natural compounds are modulatory rather than preventive.

We are very grateful for your constrictive comment. In accordance with your comments, we have changed the title in the revised manuscript to actually reflect the contents of our review.

#2. Genes encoding MyHC proteins are more commonly abbreviated as MYH and not as MyHC. I therefore suggest to the authors that they should use the commonly accepted names of these genes (in italics) MYH1, MYH2,…

We apologize for the confused abbreviations. According to your comment, we have replaced these at the table 1 in the revised manuscript.

#3. Since the focus is on maxillofacial region, a short comment regarding the sciatic nerve denervation model and atrophy/sarcopenia in the maxillofacial region would be beneficial for the paper. For instance, is the denervation model perhaps (at least indirectly) relevant for diseases, such as amyotrophic lateral sclerosis, which affects the upper and lower motor neuron and causes problems with swallowing (in addition to problems with movement)?

Thank you for your great support to improve our manuscript. In accordance with your comments, we have added the sentences which related maxillofacial region and sciatic nerve denervation model (at Line 271 - 273) in the revised manuscript.

#4. I suggest to the authors to say “myokines are secreted” instead of “myokine is secreted”. (Myokines are usually referred to as a group of molecules.)

In accordance with your comments, we have changed the sentences (at Line 203) in the revised manuscript.

#5. If there are no data regarding the effects of natural compounds on muscle atrophy/sarcopenia in the maxillofacial region, the authors could mention directly that this aspect should be studied in the future. This is relevant also because persons with swallowing problems may have insufficient intake of macro- and micronutrients, such as vitamins, which may contribute to the loss of muscles. This would be a nice conclusion of this review as well as a perspective for future research.

We are grateful for your very constrictive comments. In accordance with your comments, we have added the sentences, no studies have reported the function of natural compounds for maxillofacial muscle atrophy and the muscle of these area connects the loss of hole body muscle through malnutrition with occlusion and/or swallowing problem (at Line 448 - 452) in the revised manuscript.

#6. Chorionic disease should be chronic disease (line 36). There are a few other minor errors like this (e.g. vitamin C have instead of vitamin C has…in line 426). I suggest a careful proofreading of the manuscript.

We apologize for any typos. According to your comment, we have been careful proofreading of our manuscript again. As a result, we have changed some words (at Line 36, 326, 435 and 439) in the revised manuscript.

We thank everyone for their comments which we believe have made this a better, more compelling and readable manuscript. We have appreciated all the comments, and consider that the manuscript is substantially improved thanks to their input.